# Past, Present, and Future of Regeneration Therapy in Oral and Periodontal Tissue: A Review

**Hwa-Sun Lee [1,2,3], Soo-Hwan Byun [2,3,4], Seoung-Won Cho [2,5] and Byoung-Eun Yang [2,3,5,*,†]**

[1]   Department of Periodontology, Hallym University College of Medicine, Hallym University Hangang Sacred Heart Hospital, Seoul 07247, Korea; periodrsunny@naver.com

[2]   Graduate School of Clinical Dentistry, Hallym University, Chuncheon 24253, Korea; purheit@daum.net (S.-H.B.); kotneicho@gmail.com (S.-W.C.)

[3]   Institute of Clinical Dentistry, Hallym University, Chuncheon 24253, Korea

[4]   Department of Oral and Maxillofacial Surgery, Hallym University College of Medicine, Hallym University Dongtan Sacred Heart Hospital, Dongtan 18450, Korea

[5]   Department of Oral and Maxillofacial Surgery, Hallym University College of Medicine, Hallym University Sacred Heart Hospital, Anyang 14066, Korea

*   Correspondence: omsyang@gmail.com or face@hallym.or.kr; Tel.:+ 82-31-380-3870

†   Current address: Department of Oral and Maxillofacial Surgery, Hallym University Sacred Heart Hospital 11, Gwanpyeong-ro 170-gil, Dongan-gu, Anyang-si 14066, Korea.

**Abstract:** Chronic periodontitis is the most common disease which induces oral tissue destruction. The goal of periodontal treatment is to reduce inflammation and regenerate the defects. As the structure of periodontium is composed of four types of different tissue (cementum, alveolar bone periodontal ligament, and gingiva), the regeneration should allow different cell proliferation in the separated spaces. Guided tissue regeneration (GTR) and guided bone regeneration (GBR) were introduced to prevent epithelial growth into the alveolar bone space. In the past, non-absorbable membranes with basic functions such as space maintenance were used with bone graft materials. Due to several limitations of the non-absorbable membranes, membranes of the second and third generation equipped with controlled absorbability, and a functional layer releasing growth factors or antimicrobials were introduced. Moreover, tissue engineering using biomaterials enabled faster and more stable tissue regeneration. The scaffold with three-dimensional structures manufactured by computer-aided design and manufacturing (CAD/CAM) showed high biocompatibility, and promoted cell infiltration and revascularization. In the future, using the cell sheath, pre-vascularizing and bioprinting techniques will be applied to the membrane to mimic the original tissue itself. The aim of the review was not only to understand the past and the present trends of GTR and GBR, but also to be used as a guide for a proper future of regeneration therapy in the oral region.

**Keywords:** guided tissue regeneration; guided bone regeneration; membrane; bone graft; tissue engineering; three-dimensional scaffold

## 1. Introduction

The health of oral and periodontal tissue significantly affects the quality of life [1]. With the increase of human lifespan, the dental approaches to prevent inflammatory disease on the periodontium are increasing. The periodontium is composed of gingiva, periodontal ligament (PDL), alveolar bone, and cementum [2]. The action of the periodontium depends upon its structural integrity and interactions among its components. However, when periodontal disease occurs, the periodontium progressively gets destroyed, and this may lead to tooth loss [3].

Periodontal treatment aims to reduce the inflammation tissues caused by bacterial plaque, to correct defects due to the periodontal disease, and to, in the end to regenerate new periodontal tissue [4,5]. The ideal goal of periodontal regeneration is to obtain new cementum with PDL fibers connected to alveolar bone [5,6]. Various surgical approaches including guided tissue regeneration (GTR) and guided bone regeneration (GBR) were introduced to regenerate tissues [5–7]. Those two concepts are defined in the Glossary of Periodontal Terms 4th Edition as follows: "Procedures attempting to regenerate lost periodontal structures through differential tissue responses. Guided bone regeneration typically refers to ridge augmentation or regenerative bone procedures; GTR typically refers to the regeneration of periodontal attachment. Barrier techniques, using materials such as expanded polytetrafluoroethylene (e-PTFE), polyglactin, polylactic acid, calcium sulfate, and collagen, are employed in the hope of excluding epithelium and the gingival corium from the root or existing bone surface in the belief that they interfere with regeneration" [8]. Conventional GTR is theoretically based on the different growth speed of gingival fibroblast and the mesenchymal cells in PDL [9]. Use of a barrier membrane is a procedure for epithelial exclusion in order to promote the healing of periodontal tissues in such a way that the original structure and function is preserved instead of repairing with junctional epithelium [9]. During the occlusive period, the cells including cementoblast, osteoblast, osteoclast, and mesenchymal cells from PDL are activated to rebuild their missing tissues [5–9].

For successful regeneration, both GTR and GBR procedures are used to achieve stability of blood clot, wound site healing, and isolation of the bone healing site from soft tissues, and to provide adequate space for bone healing [10]. The "PASS principle" suggests four biological principles necessary for the bone regeneration, which are (1) primary wound closure to ensure uninterrupted healing; (2) angiogenesis to provide blood and nutrient supply, as well as delivery of pro-healing cell types; (3) space maintenance for new bone growth while preventing soft tissue in-growth; and (4) stability of wound to include blood clot formation [11]. Furthermore, an additional five principles from a surgeon's perspective are suggested as (1) the appropriate and adequate membrane must be chosen; (2) promotion of healing of primary soft tissues; (3) primary closure of the membrane when possible; (4) stabilization of the membrane at the adjacent bone; (5) sufficient long-term healing [12]. There are many advances in regeneration therapy in periodontal tissues, but only the developments of membranes and grafting materials are reviewed in this article.

## 2. Conventional GTR and GBR

### 2.1. Membrane

In GTR and GBR treatment, the membrane plays a critical role, which is the reason why various membranes are studied in different regenerative treatments. The ideal characteristics of a membrane are not always defined as reasonable absorbability or sufficient stiffness in clinical situations. However, there are some essential traits of a barrier membrane [4,5,13]. The membranes should have (1) biocompatibility to allow integration with the host tissue without eliciting inflammatory responses or immune response, (2) cell occlusiveness, which excludes undesirable cell types, (3) ability for tissue integration with a proper degradation profile to match that of new tissue formation, and (4) adequate mechanical properties to avoid the collapse of membrane and place on the bone defect [13]. The barrier membranes used in GTR and GBR are usually divided based on their degradation characteristics: absorbable and non-absorbable membranes. In this study, they are categorized based on their development period: first generation (non-absorbable), second generation (absorbable), and third generation (tissue-engineering membrane).

### 2.1.1. First-Generation Membranes

The first-generation membranes aimed to achieve mechanical properties with a minimal toxic effect on adjacent tissue. At that time, the occlusive property was of the primary focus. In 1969,

e-PTFE was developed and became a standard material in the 1990s. It has a double layer with pores between 5 and 20 microns. On one side is a 1-mm-thick open microstructure with 90% porosity, which retards the epithelial growth, and, on the other side, is a 0.15-mm-thick membrane with 30% porosity, which makes space for new bone [14]. The efficacy of e-PTFE was proven in many studies [14]. However, a higher exposure rate due to the high porosity and the additional surgery needed to remove it were its significant drawbacks. To overcome the limits of e-PTFE, high-density PTFE membrane (d-PTFE) with a smaller pore size (less than 0.3 microns) was developed in 1993 [15]. The feature of the adjusted transparency resulted in proper bone regeneration even in exposed cases, and no tissue integration into the membrane made it easy to be removed [14]. However, d-PTFE still had the limitation of collapse into the defect. Since e-PTFE and d-PTFE membranes showed low mechanical rigidity, titanium-reinforced e-PTFE and d-PTFE membranes were developed. The embedded titanium framework could be bent to fit on the alveolar defect. It was efficient for space maintenance in severe bony defects [14]. In several studies, Ti-reinforced e-PTFE showed a higher space maintaining ability and better stability than e-PTFE [16,17]. In a case which recommends proper mechanical support such as vertical bone augmentation, titanium (Ti) mesh is recommended. Ti mesh has high rigidity, elasticity, stability, and plasticity, and its holes allow good blood supply directly from the periosteum to the surrounding tissues [15]. However, stiff Ti mesh tends to be exposed more often, and it is difficult to remove it due to the bone tack or other fixation devices [13].

### 2.1.2. Second-Generation Membranes

In order to eliminate the necessity for secondary surgery, an absorbable membrane was suggested as a barrier membrane for GTR and GBR. The absorbable membranes are divided into two groups based on their origin: natural membranes and synthetic membranes.

Natural membranes mainly consist of collagen or chitosan from animal sources. Tissue-derived collagen-based membranes are made from human skin (Alloderm®, LifeCell, Branchburg, NJ, USA), bovine achilles tendon (Cytoplast® RTM Collagen, Franklin Lakes, NJ, USA), or porcine skin (Bio-Gide®, Geistlich, Shirley, NY, USA) [18,19]. Collagen has many biological activities such as hemostasis, an attraction of periodontal ligament and gingival fibroblast cells, augmentation of the soft tissue, biocompatibility, biodegradability, and cell affinity. Most of the commercial collagen membranes are developed from type I collagen or a combination of type I and type III collagen [14]. Collagen-based membranes showed inferior performance in vivo, as the membrane starts to degrade [20]. Additionally, disease transmission and ethical or religious problems can be possible risks. Biomechanical properties and stability of the collagen matrix can be enhanced by physical and chemical cross-linking, ultraviolet light, hexamethylene diisocyanate (HMDIC), glutaraldehyde (GA), diphenyl-phosphoryl azide (DPPA), formaldehyde (FA) plus irradiation, and genipin (Gp) [21,22]. The addition of a natural cross-linking agent, Gp, into the AlloDerm® rehydration protocol affects the mechanical properties and stability of the collagen matrix [21]. A significant enhancement in tensile strength compared to control was observed when Gp exposure time was increased from 30 min to 6 h. Also, other studies showed that cross-linking is associated with prolonged biodegradation, reduced epithelial migration, decreased tissue integration, and decreased vascularization [23]. Also, the membrane made from silk fibroin showed a biocompatible reaction with osteoblastic-like MG63 cells and it could be an alternative barrier membrane for GBR [24].

Polymeric membranes are either based on polyesters such as polyglycolic acid (PGA), polylactic acid (PLA), poly ε-caprolactone (PCL), their copolymers, and tissue-derived collagens [5,25]. These materials are biocompatible, biodegradable, and easier to handle than e-PTFE membranes. In general, the membrane should maintain its function for more than 4–6 weeks for successful periodontal regeneration [26]. However, they are not inert, since some tissue reactions may be expected during degradation. Although the initial tensile strength of membranes was high (12–14 MPa), it completely decreased below 1 MPa after 14 days of exposure [27]. There is also variability and lack of control over the rate of membrane absorption, which is influenced by factors such as the local pH and material

composition [27]. The polymeric membrane has other drawbacks, including an inflammatory reaction due to macrophages and leukocytes around the membrane during absorption [28]. These membranes are processed by melting or solvent casting/particulate leaching and phase inversion [27].

### 2.1.3. Third-Generation Membranes

By reviewing the previous absorbable and non-absorbable membranes, interests should arise in developing a new membrane which has a more advanced role as a barrier membrane and has an additional function such as releasing beneficial agents such as antibiotics, growth factors, and adhesion factors at the wound. The substance releasing membrane should have a proper releasing time according to the environment of the graft site [29].

### *2.2. Membranes with a Functional Layer*

### 2.2.1. Membranes Releasing Antimicrobial Agent

The control of bacterial inflammation in the periodontium is essential not only at the nonsurgical stage, but also at the surgical stage for successful regeneration [3]. Antibacterial agent (25 wt.% metronidazole benzoate (MET)) can be added into the layer near the epithelial tissue, and it prevents bacterial growth and biofilm formation, as determined by SEM images compared to the control group [13]. Another agent like tetracycline hydrochloride (TCH), which is also valid for periodontal pathogens, can be annexed to the membranes with PLA and poly(ethylene-*co*-vinyl acetate) (PEVA). The release of TCH was highest in the PEVA scaffold [30]. As tetracycline has an elongated release period in collagen membranes than other antibiotics, it could be applied where the long-term usage of the membrane is required. Furthermore, these antibiotic-coated membranes did not exhibit any cytotoxic effects [30]. PCL- and PLA-based barrier membranes were also loaded with MET for anti-infective GTR solutions [31].

### 2.2.2. Membranes Releasing Growth Factors

The essential factors for periodontal regeneration depend on the interactions among the scaffold material and growth factors, cells, and blood supply [29]. In particular, growth factors have an essential role in the healing process. They influence tissue repair including angiogenesis, chemotaxis, and cell proliferation, and they modulate the synthesis and degradation of extracellular matrix proteins. Several bioactive molecules such as platelet-derived growth factor (PDGF), basic fibroblast growth factor (b-FGF), tumor growth factor (TGF), bone morphogenic protein (BMP), and enamel matrix derivatives (EMD) showed positive outcomes in stimulating periodontal regeneration [32]. In several studies, PDGF-BB-loaded PLLA membranes showed an enhanced regeneration procedure and proliferation of PDL than the control group [32,33]. Other studies on b-FGF with collagen sponge, TGF-beta with PLLA membrane, and recombinant BMP-2 with hybrid alginate/nanofiber mesh showed effective regeneration in defects [34]. EMD is assumed to elicit the new attachment formation upon inserting collagen fibers between new alveolar bone and new cementum [34]. The use of EMD affects the gene expression, protein synthesis, and stimulation of osteoblast and PDL cells [34].

### 2.2.3. Platelet-Rich Fibrin (PRF) Membrane

As other growth factors showed improved results, the autologous growth factors in platelets drew attention. Within platelet granules, many growth factors are included which stimulate soft-tissue healing [35]. PRF membrane is easily obtained from centrifuged blood (3000 rpm, 10 min) and is a cost-effective alternative to other collagen membranes.

### 2.2.4. Membranes with Calcium Phosphate

Nanosized hydroxyapatite (HA) particles were incorporated into the membrane to improve biocompatibility and osteoconductivity, and the result was significant [36]. This was induced by

the action of HA for early cell differentiation. In one study, a nanofibrous bone-mimicking scaffold was electrospun from a mixture of PCL, collagen I, and HA nanoparticles with a dry-weight ratio of 50/30/20, respectively (PCL/col/HA). The cytocompatibility of it was compared with three other scaffold formulations: 100% PCL, 100% collagen I, and a bi-component scaffold containing 80% PCL/20% HA (PCL/HA). The result showed more rapid cell spreading, and significantly greater cell proliferation. MSCs were proliferated on the tri-component membrane. The cells seeded onto PCL/col/HA scaffolds showed markedly increased levels of phosphorylated focal adhesion kinase (FAK), a marker of integrin activation and a signaling molecule known to be necessary for directing cell survival and osteoblastic differentiation [36].

### 2.2.5. Amniotic Membrane

The amniotic membrane (AM) is developed from extra-embryonic tissue and has an epithelial monolayer, a thick basement membrane, and a vascular stroma (mainly collagen). Even though the AM has no blood vessel or nerve, it can transfer nutrients by diffusion through amniotic fluid. As the basement membrane of amnion is similar to that of gingiva, AM was applied to GTR as a membrane containing growth factors [37]. In a clinical study using allograft bone material with an AM for GTR therapy, the improvement of clinical attachment and reduction of pocket depth was obtained during a 12-month follow-up [38]. Also, AM could be used a cell scaffold in tissue engineering due to its biocompatibility. The epithelial cells of AM secrete collagen type III and IV and non-collagenous glycoprotein [37,38]. Perlecan is a critical component of basement membrane which is involved in the binding of growth factors and interacts with various cell adhesion molecules [39]. From these features, AM could be a possible scaffold for a suitable membrane for cell seeding which has additional anti-inflammatory and anti-scarring characteristics, as well as low immunogenicity and proper mechanical properties [37].

### 2.2.6. Cell Transferred Membrane

Owing to the development of tissue engineering techniques, a cell containing a membrane can be a tool for cell transfer [40]. From previous studies, it is clear that the induction of capillary system and progenitor cells into the defect area could be the key to regeneration [11]. Even after inducing progenitor cells, the progenitor cell should be differentiated to the target cell, and it takes time to start regeneration. Therefore, the idea of transplanting specific cells into the membrane for cell delivery to the area was suggested upon this basis [40]. The method of cell transfer can be various and, recently, culturing the cells on the adhesive area and then transferring them to the surface of a three-dimensional (3D) scaffold became a standard procedure. The cell sheets can be made from various cell types such as fibroblasts, mesenchymal stem cells (MSC), osteoblasts, and endothelial cells [40–43]. Also, more than one cell type can be transplanted simultaneously by the same method [42]. Especially in the GBR area, the carotid artery endothelial cell and osteoblast could be an ideal combination. However, in GTR usage, the space for PDL should be appropriately considered.

### 2.3. Membranes with Zone-Dependent Bioactivity

### 2.3.1. Electrospinning (E-Spinning) Membrane

The e-spinning apparatus includes a polymer solution/melt in a syringe, charged through a high-voltage supply, and a grounded plate positioned at a predetermined distance from the tip of the needle [29]. The potential difference overcomes the surface tension of the fluid droplet at the tip of the metal needle, which in turn results in the formation of the so-called Taylor cone. The fluid jet experiences whipping instabilities and tends to dry and form fibers with an average diameter ranging from several microns to tens of nanometers [44]. The controlling parameters including voltage, distance from tip to the collector, collector type (rotating or static), solution properties, and flow rate present a significant influence on fiber formation and morphology [44,45]. The result can be composed of

biocompatible and degradable natural or synthetic polymers or blends, and resembles the arrangement of the native ECM. The three-dimensional (3D) structure made by the e-spun process showed improved hydrophilicity and wettability, which stimulates cell and ECM interaction and increases the cell proliferation and attachment by providing physical and chemical stimuli to cells [46]. By repeating e-spinning, multilayers can be formed, and this process is used to produce membranes with different layers and different materials such as the combination of polymers and natural proteins [47].

### 2.3.2. Multilayer Membranes with a Functionally Graded Structure

For true regeneration during GTR and GBR, the membrane needs to modulate new periodontium by enhancing alveolar bone growth while preventing the downgrowth of gingival tissue [48]. By using the e-spinning technique, each layer can be fabricated with different materials with different features [46–48]. The functionally graded membrane (FGM) consists of a core layer (CL) and two functional surface layers (SL), each facing bone (nano-HA (hydroxyapatite)) and epithelial (MET (metronidazole)) tissue. The CL is composed of a poly (D,L-lactide-*co*-caprolactone) (PLCL) layer surrounded by two composite layers composed of a gelatin/polymer ternary blend [29]. The features of CL, such as favorable elasticity and high tensile strength under hydration (8.7 MPa), make the membrane stable during the regeneration period [48]. However, the other two SL layers showed decreased tensile strength under dry or hydrated conditions [48].

The schematic structure and degradation process are described in Figure 1. The 3D structure is formed with different nanofibers in various directions according to their manufacturing condition, and they have 3D space within their structure for revascularization and cell infiltration. The structure can also be understood from its sagittal view, and has two surface layers and one core layer [29]. These structural features induce an appropriate degradation process started by hydration and revascularization, and when some degree of volume loss of scaffold is obtained, progenitor cell invasion follows. After cells progressively differentiate into tissue-specific cells, the membrane itself is metabolized to make space for newly formed tissues. The phase of degradation of 3D scaffolds is important due to the harmony of new tissue formation.

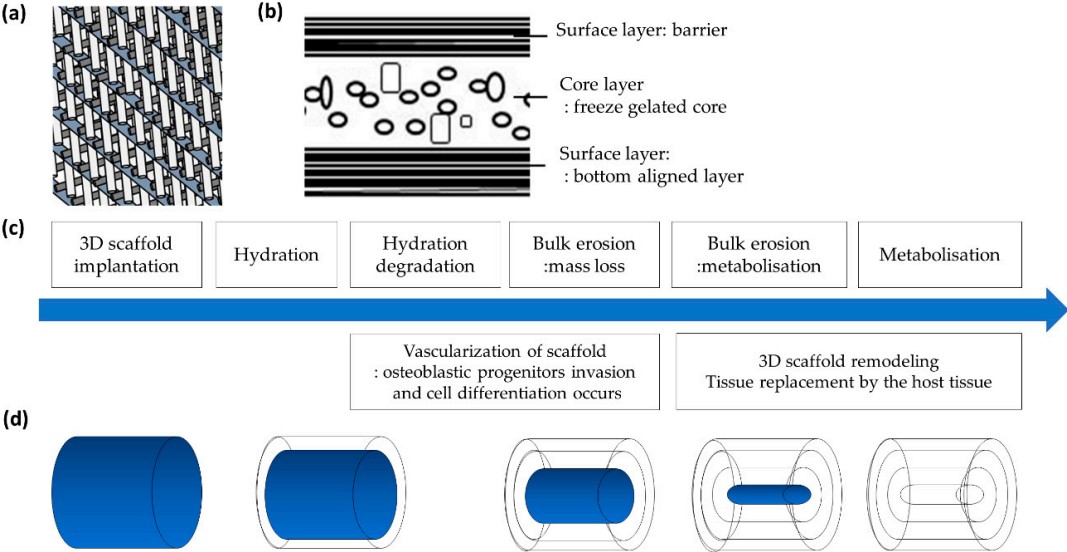

**Figure 1.** The schematic structure and degradation process of three-dimensional (3D) scaffolds. (**a**) Schematic structure of 3D scaffold with three different nanofibers; they form an inner space with a different biological feature for zone-dependent reaction during the regeneration period. (**b**) Sectional layers of a functionally graded membrane; two surface layers and one core layer with a different function are shown. (**c**) The degradation process of 3D scaffolds related to biologic reactions within a tissue. (**d**) The change of fibers consisting of a 3D structure according to the degradation procedure; space is not a hollow space but newly regenerated tissue.

### 2.4. Bone Graft Materials

The periodontal regenerative treatment aims for alveolar bone regeneration with PDL space around the root. Bone tissue can repair itself; however, in the case of severe destruction, it cannot be recovered to the amount of the original structure. The bone graft material is applied in that case to induce bone regeneration [10,14]. The ideal bone graft material may vary based on defect size, tissue viability, shape and volume, cost, biological characteristics, etc. [49]. The characteristics of the bone materials are divided based on their viability. Osteogenesis is the ability to produce new bone by osteoblasts which came from the progenitor cells in host or graft material [50]. Osteoinduction is the ability to induce bone formation by secreting growth factors (TGF, BMP, PDGF, and FGF) to surrounding tissue to stimulate osteoblast differentiation from the stem cell [49]. Osteoconduction is the function of mechanical support which supports blood vessel ingrowth [51]. Among all types of bone graft materials, only the autograft has all types of viability and allografts, and xenografts have one or two viabilities [49–51].

#### 2.4.1. Histologic Type of Bone Graft Material

Normal bone structure is composed of outer cortical bone and inner cancellous bone. Cortical bone has a higher mineral component (80–90%) than cancellous bone (15–25%), and it shows higher strength and stiffness. In contrast, cancellous bone has a more metabolic function in bone-marrow space which can exchange cells and signals [51]. Cancellous bone is typically used to fill small defects to enhance bone formation. It stimulates faster revascularization and bone in-growth, but it cannot achieve high strength. The cortico-cancellous block is used in cases where the support of a structure and osteogenesis is required, such as in articular surface reconstructions [49,50].

#### 2.4.2. Type of Bone Graft Material

(i)     Autografts

Autogenous bone grafts are bone tissue which is harvested from one site of the body and implanted into other sites in the same person. The grafts can be cortical or cancellous bone or cortico-cancellous grafts. Fresh autogenous bone contains the surviving cells and osteoinductive proteins including BMPs. They are the best materials due to lacking immunogenicity and fast vascularization [52]. Possible donor sites can be the distal radius, tibia, iliac crest, mandibular ramus, and maxillary tuberosity [53]. Autogenous bone grafts in periodontal regeneration are a very cheap and effective option, but there are some limitations such as fast resorption, donor site morbidity, and limited amount [52] (Table 1).

(ii)     Allografts

Allografts are extracted from one person and implanted into another person from the same species. Due to several limitations of autografts, allografts are used clinically as a common alternative to autografts. They can be achieved from cadavers or living persons [54]. Cancellous allografts provide minimal to no structural strength, mild-to-moderate osteoconductivity, and mild osteoinductive properties. In contrast, cortical allografts show good strength but little osteoinductivity [49]. Freeze-dried and fresh-frozen bone allografts induce more prompt graft incorporation, vascularization, and bone regeneration than fresh bone allograft. Freeze-drying of bone allografts produces a safer bone graft regarding the decrease in the risk of immunologic responses and transmission of diseases [55,56]. As the osteoblast and BMP are denatured through the processing course of allografts, freeze-dried bone allografts showed delayed bone incorporation and revascularization than autografts. The freeze-drying method also decreases the mechanical strength of the allografts, and the cost of processing the allografts is high. The demineralized allografts possess osteoconductive and osteoinductive properties, and they vascularize fast [55].

(iii)   Xenografts

Xenografts are bone tissue harvested from different species, commonly derived from coral, bovine, and porcine sources, consisting of hydroxyapatite bone mineral. In the study comparing different species (canine bovine, porcine, and coral graft), bovine bone xenografts had organic substances extracted, and had a non-antigenic, natural porous matrix, identical to the mineral phase of bone tissue [57]; they were shown to have high osteoconductive properties and a meager absorption rate. Also, another study reported that an inorganic bovine bone has no osteoinductivity and its granular form makes it difficult to be fixed on surgical sites. Moreover, the bovine xenograft is non-absorbable in vivo [58].

**Table 1.** Advantages and disadvantages according to the types of grafts.

| Types of Grafts | Advantage | Disadvantage |
| --- | --- | --- |
| Autograft | Osteogenesis: containing live cells<br>Osteoinductive: having BMP and other growth factors<br>Osteoconductive<br>Lack of immunity<br>No disease transmission<br>Cost-effective | Donor site morbidity due to harvesting<br>Pain<br>Limited donor site: limited amount |
| Allograft | No morbidity of donor site<br>Unlimited amount<br>Osteoinductive, osteoconductive<br>Various mineral composition: cortical, cortico-cancellous, cancellous<br>Various form: powder, cancellous cubes, cortical chips/fresh, fresh-frozen, freeze-dried/mineralized, demineralized | No osteogenesis: no live cell inclusion<br>Disease transmission: viral or bacterial, 12.9–13.3%<br>High cost<br>Dependent on donor's bone state: age<br>Ethical problem |
| Xenograft | No morbidity of donor site<br>Unlimited amount<br>Osteoconductive | No osteogenesis<br>No osteoinduction<br>Disease transmission<br>Non-resorbable in vivo<br>Ethical problem |

## 3. New GTR and GBR Using Tissue Engineering

### 3.1. Periodontal Tissue Engineering (TE)

Previously regenerated tissues via conventional GTR or GBR procedures using graft materials and membranes showed an insufficient mechanical strength, unstable volumetric stability under an external force and limitations for application in a case with an extensive range of tissue defects [4,5]. To overcome these limitations, tissue engineering (TE) was introduced as regenerative medicine. TE is a technique which fabricates tissues outside of the body and implants them into the body to regenerate the lost target tissues [59]. The classic TE paradigm includes four essential requirements: (1) scaffold, i.e., biomaterials which provide space for new cell ingrowth; (2) biological agent, i.e., appropriate regulatory signals; (3) cells, i.e., responsive progenitor cells; and (4) blood supply [60,61]. The scaffold can be either acellular or cellular, whereby the former provides adequate space to recruit local stem cells and osteoprogenitor cells, and the latter incorporates the involved stem cells before the implantation [62]. The fabrication of a successful scaffold plays a crucial role in TE; thus, new technologies focus on 3D scaffolds [61]. Therefore, in this review, we discuss scaffolds, while other elements such as cells and biological agents were discussed in another review [63].

### 3.2. Scaffold-Based Tissue Engineering

The scaffolds act as a mechanical structure to support cell infiltration and revascularization. Their design should consider biochemical properties and degradation kinetics to mimic original tissue biology. The external and internal geometry can be referred to as a three-dimensional (3D) scaffold which promotes cell adhesion, proliferation, differentiation, and bone mineral deposition [64,65].

In designing 3D scaffolds, the following concerns must be considered: (1) proper architecture such as porosity, pore size, and interconnectivity; (2) mechanical properties to bear the external force during regeneration; (3) physical properties like hydrophilicity, roughness, and surface topography; (4) biocompatibility like cell affinity; (5) biodegradability in harmony with bone maturation; (6) sterilizability without loss of bioactivity; and (7) controlled deliverability of bioactive molecules or drugs [64,65]. Human cancellous bone has a total porosity from 30% to 90%, and cell interaction occurs within the voids [66]. Several preclinical and clinical studies suggest that scaffolds for alveolar bone regeneration should have 70% overall porosity [66,67]. When it comes to the pore diameter, a range between 150 μm to 500 μm promotes vascularization and infiltration of new cells and tissues without compromising mechanical strength [66]. Not only the presence of pores, but also the interconnection between them is crucial in cell growth within the network and prevention of core necrosis [68]. The physical properties are closely related to biomaterials utilized to make the scaffold [64,65]. The scaffold should have biocompatibility, bioactivity, and non-cytotoxic effects [68]. Also, adequate hydrophilicity and roughness, and specific surface nanotopography should be developed to replicate the natural process of bone regeneration [69]. Nanotopography increases the surface area, surface-to-volume ratio, and surface roughness [70]. Finally, the control of the scaffold degradation process is the most critical key in successful tissue regeneration. In the early days, even tissue ingrowth was not actual regeneration; however, the sign of direct immature tissue filling was considered a sign of scaffold degradation. Thus, scaffold degradation was faster than tissue remodeling and maturation, which made proper regeneration fail [60]. The onset of scaffold degradation should be followed after tissue remodeling within the scaffold at least once in the natural remodeling cycle [71]. Recently, the concept of scaffolds changed to have sufficiently elongated mechanical strength to bear collapse during tissue maturation. In order to harmonize with dentoalveolar remodeling, scaffold degradation within 5–6 months is considered appropriate [71]. Using this paradigm, a multiphasic scaffold with a different structure and chemical composition was introduced for periodontal regeneration [64]. As the periodontium is a complex of soft and hard tissues, different features of each scaffold layer can induce functionally oriented PDL into the alveolar bone and the cementum over time [64]. A 0.0250-mm-thick space to mimic PDL space and to prevent tooth ankylosis was tried in one study and showed positive results [72].

### 3.3. Biomaterial Scaffold for Bone Cell Infiltration

As reported before, biomaterials influence the overall properties of the scaffold [64]. The individual characteristics of biomaterials should be compared to find their optimal for the ideal scaffold (Table 2).

#### 3.3.1. Natural Polymers: Collagen, Alginate, Chitosan, and Hyaluronic Acid

Natural polymers were adopted as the first biomaterials because of their high biocompatibility, proper cell recognition, enhanced cellular interaction with surrounding tissue, and hydrophilicity [73]. Based on these properties, they were investigated as hydrogels containing cells inside and showed successful results [73,74]. Collagen is the most abundant protein found in bone, and it provides biocompatibility and structural stability to many tissues [75]. The collagen matrices promote cell adhesion, proliferation, and osteogenic differentiation of bone-marrow stromal cells in vitro [75]. The limitations of collagen, such as low mechanical strength, were improved by modifying collagen cross-linking [76–78]. Similarly, the denatured form of collagen, called gelatin, enhances osteoblast

adhesion, migration, and mineralization based on biological groups [75]. Alginate is a natural material derived from brown algae which forms a gel in contact with water [79]. It is a polysaccharide and is highly processible to make various forms of scaffolds, and it is easy to encapsulate living cells [79]. The viscosity and the porosity of alginate enable cellular immobilization, integration, and the extended release of factors and cells from the scaffold [80]. To overcome the low mechanical strength, alginate is often combined with other compounds such as chitosan, gelatin, and hydroxyapatite to improve osteoconductivity [80,81]. Another essential feature of alginate is its factor-releasing effect to enhance cell affinity and angiogenesis [82,83]. In the polysaccharide group, chitosan is another popular biomaterial based on its antibacterial and antifungal effects [84]. When applied in a scaffold, chitosan accelerates wound healing by avoiding infection of the operation site and preventing exposure of the wound. Hyaluronic acid is another natural polymer and is an essential element in wound healing by aggregating glycosaminoglycans (GAG) in the joint capsule [85]. Hyaluronic acid combines with other compounds to enhance mechanical properties and often contains growth factors for additional regenerative potential [86,87]. Despite their advantageous biological properties, natural polymers showed a lack of bioactivity for promoting bone tissue regeneration, as well as a rapid degradation rate related to low mechanical strength [79,88]. To overcome those limitations, natural polymers are usually combined with bioactive materials (i.e., bioceramics) or mechanically strong materials (i.e., synthetic polymers or metals) [89].

### 3.3.2. Synthetic Polymers: Polycaprolactone (PCL), Polyethylene Glycol (PEG), Polylactic Acid (PLA), Polyglycolic Acid (PGA), and Polylactic-*co*-Glycolic Acid (PLGA)

Synthetic polymers are utilized for their low cost and their ability to be produced in large quantities with a long shelf life [90]. PCL is a biodegradable compound which has mechanical strength and manufacturability for bone TE. With the help of many processes such as photopolymerization and 3D printing, PCL can have a porous structure [91], and this structure can be used to seed mesenchymal cells and growth factors to improve regeneration [92]. PCL also has biocompatibility and a remarkably slow degradation rate [93]. However, PCL is hydrophobic, which is responsible for low cell affinity and poor cellular response [94]. Similar to PCL, PLA and PLGA are hydrophobic. PEG has low toxicity, in addition to being hydrophilic and soluble. However, it shows a higher degradation rate than PCL [95]. Synthetic polymers degrade via hydrolysis within the interior part of biomaterial, resulting in an empty shell formation [96]. These features make them useful for bone graft materials, but not suitable for a drug-delivery system. Also, their acidic byproducts during degradation induce tissue necrosis and resultant exposure of the scaffold [61]. Therefore, bioceramics are usually combined with synthetic polymers to enhance bioactivity and to neutralize the acidic byproducts [97].

### 3.3.3. Bioceramics: Hydroxyapatite (HA), Bioactive Glass, β-Tricalcium Phosphate (β-TCP)

Bioceramics are inorganic biomaterials which are well documented as bone fillers in dental applications [98]. Calcium phosphate bioceramics are moldable, easy to handle, and they get hardened in situ, which enables them to adapt in complex defects [99]. Also, they have bioactivity, excellent biocompatibility, hydrophilicity, similarity to native bone inorganic components, osteoconductivity, and potential osteoinductivity [99] This potential activity is possible with a surface of bioceramics which absorbs and exhibits osteoinductive factors or via the gradual release of calcium and phosphate ions into nearby tissue, stimulating the differentiation of osteoblasts [100]. HA is the most common ceramic of TE and improves the adhesion and proliferation of osteoblasts with bone-like mineral components [100]. However, the crystalline form of HA showed a slow degradation rate, and it is relatively brittle when bearing weight [101]. Methods to improve the mechanics, such as adding other compounds, sintering, and using amorphous HA which has a faster degradation rate, were introduced [101–103]. Bioactive glass is a silicon oxide with substituted calcium [98]. It forms a calcium phosphate layer in the surface when exposed to body fluid, and its biocompatibility and surface pores allow tissue in-growth [104]. β-TCP is widely used due to its ability to form a strong bone–calcium

phosphate bond and its faster degradation rate [105]. Interestingly, when β-TCP is combined with HA, a mixture termed biphasic calcium phosphate (BCP) is produced [106]. In comparison to other calcium phosphate ceramics, BCP has significant advantages including controlled bioactivity and stability, while promoting bone in-growth, especially in large bone defects; BCP also has a higher degradation rate than HA, but a slower one than β-TCP [106]. Even bioceramics have ideal properties for their use as scaffold materials, and they are very brittle due to their high stiffness and low flexibility [107]. Thus, their combination with supporting materials such as PLLA and PLGA was tried and the results showed improved mechanical properties and osteogenic potential [108].

**Table 2.** Advantages and disadvantages according to biomaterials for the scaffold.

| Types of Grafts | Advantage | Disadvantage |
| --- | --- | --- |
| **Natural polymers**<br>Collagen<br>Alginate<br>Chitosan<br>Hyaluronic acid | High biocompatibility<br>Enhanced cellular<br>Interaction<br>Hydrophilicity<br>Antibacterial effect<br>Cell/drug containing | Lack of bioactivity<br>Rapid degradation rate<br>Low mechanical strength |
| **Synthetic polymers**<br>Polycaprolactone (PCL)<br>Polyethylene glycol (PEG)<br>Polylactic acid (PLA)<br>Polyglycolic acid (PGA)<br>Poly(lactic-*co*-glycolic) acid (PLGA) | Mechanical strength<br>Can be processed variously<br>Able to seed mesenchymal<br>cells/growth factors | Slow degradation rate<br>Hydrophobic (PCL)<br>Low cell affinity<br>Poor cellular response<br>Not suitable for a drug-delivery system<br>Acidic byproducts |
| **Bioceramics**<br>Hydroxyapatite (HA)<br>Bioactive glass<br>β-tricalcium phosphate (β-TCP) | Easy to handle<br>Bioactivity<br>Good biocompatibility<br>Hydrophilicity<br>Similar inorganic components<br>Osteoconductivity<br>Potential osteoinductivity | Very brittle<br>High stiffness<br>Low flexibility |

### 3.4. Three-Dimensional (3D) Scaffold Fabrication Techniques

Previously, conventional methods including particle leaching, gas foaming, freeze drying, phase separation, fiber bonding, melt molding, and solution casting were employed for 3D scaffold fabrication [109]. As those methods revealed heterogeneities in pore size, porosity, and architecture, a new technique called solid-freeform fabrication (SFF), which is known as rapid prototyping (RP), was developed. SFF can fabricate scaffolds with precise external shape, internal morphology, and reproducible 3D architecture [110]. Several additive manufacture processes can be applied to build a complex structure by 3D printing. Those include laser-assisted printing (e.g., selective laser sintering (SLS) and stereolithography (SLA)), inkjet printing, and extrusion-based printing (e.g., fused deposition modeling (FDM)). Each procedure has their indications according to the different features and viscosity of biomaterials; materials with low viscosity can be adapted to inkjet printing, while only thermoplastic biomaterials can be used in extrusion-based printing, and a wide range of viscosities can be printed by laser-assisted printing [111]. Also, they can utilize living cells in printing, and the printed cells can be kept in a hydrogel scaffold made with natural or synthetic polymers [81,94,111]. All of these technologies use the computer-aided design (CAD) and computer-aided manufacturing (CAM) software or digital images for the design [112,113]. CAD models are produced using direct computed tomography (CT) scans of patient-specific bone defects, which could help in regenerating extensive or complex forms [113]. Few studies using image-based 3D-printed scaffolds showed promising results in practice [72,114,115]. Extrusion-based printing is the most widely used method in periodontal regeneration. This technique includes the controlled extrusion of a material through a printer head onto the collector, and the dimension of the filament is adjusted by controlling printing conditions such as temperature, extrusion rate, and velocity of the collector [112,114]. Its features include

temperature-controlled material handling, a dispensing system, and an optional light source and piezoelectric humidifier [113]. FDM is a common extrusion method and thermoplastic material is fed from the filament coil and inserted into the heated nozzle head for deposition of semi-molten state polymer [115]. Another scaffold fabrication technique introduced in periodontal regeneration is electrospinning which consists of a polymer syringe, a syringe pump, a high-voltage supply, and a collector plate [113]. It can be classified into solution electrospinning (SE), which uses polymer solution, and melt electrospinning (ME), which uses polymer melts [116]. In SE, polymer solution is extruded in a whipping and oscillating motion, which enables forming micro- or nanofibers. In contrast, ME uses high-viscosity polymer which enables direct writing without being influenced by voltage instabilities [116]. Electrospinning can also be categorized by the number of fiber layers used: co-axial and tri-axial [117,118]. For co-axial, two syringes are separated inside the jet, and inner fluid and outer fluid are formed as a core–shell structure on the collector under high voltage. Tri-axial electrospinning uses a three-polymer solution: core liquid, inner shell liquid, and outer shell liquid. Triaxial fibers can be modulated to have different hydrophobicity and mechanical strength [118].

*3.5. Clinical Application of 3D Scaffolds*

By studying the various biomaterials and technologies for the fabrication of scaffolds, new scaffolds were adapted to induce ideal periodontal regeneration.

3.5.1. Scaffolds for GTR

To be applied in GTR, the scaffold must function both as a grafting material and a membrane. Thus, the scaffold should have the mechanical strength enough to maintain a space during tissue regeneration [110]. This is especially important in GTR, as the periodontal bacteria always impede flap closure and regeneration of tissue, whereby natural polymers which have antibacterial properties such as chitosan could be a good selection as a scaffold material [84]. Gelatin is another recommendable biomaterial for GTR due to its biological cell affinity [75]. Usually, natural polymers are combined with other synthetic polymers and processed by fiber-guiding a 3D PCL/HA scaffold system [72,115]. The concept of compartmentalization is obtained using a customized 3D scaffold which can achieve regeneration of PDL, cementum, and alveolar bone [110]. The biphasic construct allows not only the regeneration of obliquely oriented periodontal fibers, cementum-like tissue, and alveolar bone, but also greater control of tissue infiltration when compared to random porous scaffolds [115].

3.5.2. Scaffolds for GBR

For application in socket preservation, and alveolar bone regeneration and augmentation, scaffolds made of bioceramics combined with mechanically strong materials are recommended. In a load-free area, collagen can be the option for mixing [100]. The scaffold should degrade within five to six months for the ideal tissue apposition, and maturation and mechanical strength should be adequate to avoid the collapse of operation sites [71]. PCL is the most common scaffold due to its high mechanical strength and a variety of manufacture techniques. However, its delayed degradation proved to inhibit the osteoconduction, and acidic byproducts can expose the flap earlier and lead to failure of the regeneration process [91,94]. The use of PCL alone is not recommended, and it should preferably be combined with bioceramics such as HA for controlling the degradation rate and neutralization of the byproduct [97]. The combination of collagen and HA is encouraged for GBR due to the compositional similarities to native bone and reasonable degradation rates [53]. Bioceramics can be applied mainly in sinus elevation and bone augmentation procedures. The scaffold replaces the need for membranes and provides better mechanical support, such as the application of 3D-printed BCP scaffolds (HA 30%, β-TCP 60%, and α-TCP 10%) grafted in sinus, which showed favorable outcomes [119]. When combined with bioceramic, the increase in degradation rate regarding the percentage of its composition should also be considered.

## 4. Future Studies for Scaffolds in GTR and GBR

Until now, various biomanufacturing techniques enabled the formation of more advanced scaffolds for regeneration therapy. Although many options were introduced, there are still some limitations due to the biomaterials themselves and their combination of the manufacturing process. To obtain an ideal 3D scaffold suitable for periodontal tissue, scaffolds with cell sheets, pre-vascularized scaffolds, and scaffolds with nano-design and bioprinting should be studied further.

The phase of tissue regeneration depends on the infiltration of mesenchymal cells from surrounding tissue and their differentiation into correctly specified cells such as osteoblasts, cementoblasts, and PDL fibroblasts. A biphasic scaffold with porous structure was already invented, and additional approaches for promoting cell reaction are expected soon. As re-vascularization is another crucial factor of fast regeneration, seeding pro-angiogenic factors into the scaffold before implantation should be discussed, in addition to other approaches such as harvesting vascular bundles for the defect or vascularizing sheets of cells before insertion [120–122]. To construct a vascularized cell sheet, human mesenchymal cells and human umbilical vascular endothelial cells are added. For example, one such sheet was combined with a β-TCP scaffold, and it showed enhanced angiogenesis and bone matrix apposition [122]. Further studies are needed to determine promising results with a controlled procedure.

More advanced regeneration targets having original biology of tissue after implantation of biomaterials. The polymers used in TE have both advantageous and disadvantageous features. Therefore, a precise coupling of synthetic polymers with a biologic component could be the future trend of biomaterials. However, the combination method is yet to be thoroughly tested. In the future, optimal manufacturing procedures with a reproducible manual should be completed. Bioprinting is a technology utilizing 3D printing to combine cells, growth factors, and biomaterials to imitate natural tissue. Generally, they use a layer-by-layer method to deposit materials such as bioinks [123]. Although they have high similarity with the original tissue, this technique should be approved by governments and the protocol should be more standardized. Finally, bioprinting could accomplish patient-customized tissue for ideal regeneration.

## 5. Conclusions

For the treatment of many patients suffering from the depletion of periodontal tissues, GTR and GBR were developed for over 30 years. At first, the bone graft materials and membranes used in GTR and GBR were studied, and the standard indications of each material were determined. Then, the studies went further to change the paradigms of GTR and GBR by using scaffolds instead of graft materials and membranes. The scaffolds can have various mechanical and biologic properties regarding their basic biomaterials and manufacturing process. Also, the structure of scaffolds became more stereoscopic, implementing 3D architectures. With the development of CAD/CAM systems and 3D printers, scaffolds can have multiphasic layers for the ideal induction of original tissue compartments. Furthermore, additional processing for the improvement of 3D scaffolds, such as adding growth factors or adopting various cells, was studied extensively. However, the newly developed scaffolds remain to be widely applied in actual clinical situations, and they still have some limitations. In the future, continuous attempts should be made to develop the optimal biomaterials for predictable patient-customized regeneration in preclinical and clinical aspects.

**Author Contributions:** Conceptualization, H.-S.L. and B.-E.Y.; methodology, H.-S.L.; investigation, H.-S.L.; resources, H.-S.L.; data curation, H.-S.L.; writing—original draft preparation, H.-S.L.; writing—review and editing, S.-H.B., S.-W.C., and B.-E.Y., visualization, H.-S.L.; supervision, S.-H.B. and B.-E.Y.; project administration, B.-E.Y.

**Funding:** This research received no external funding.

**Conflicts of Interest:** The authors declare no conflict of interest.

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
