# Peer review of "Past, Present, and Future of Regeneration Therapy in Oral and Periodontal Tissue: A Review"

_applsci, doi:10.3390/app9061046_

Round 1

Reviewer 1 Report

Manuscript title: Past, Present and Future of Regeneration Therapy in Oral and

Periodontal Tissue:

The authors review that past and present trend of GTR and GBR in oral region.

The topic and the results of this study are interesting. The reviewer thinks the manuscript can be published after the minor revision of below things.

 The authors describe GTR and GBR. Clinically the regeneration therapy using growth factor, such as EMD, PDGF, PRP is often applied. That information should be added.

Author Response

Response to Reviewer 1 Comments

Point 1: The authors review that past and present trend of GTR and GBR in oral region.

The topic and the results of this study are interesting. The reviewer thinks the manuscript can be published after the minor revision of below things.

The authors describe GTR and GBR. Clinically the regeneration therapy using growth factor, such as EMD, PDGF, PRP is often applied. That information should be added.

Response 1:

Dear reviewer 1,

Thank you for your kind review.

We have reviewed a great number of journals on various types of regeneration therapy to write this literature. However, since too many perspectives were discussed in those journals, we have decided to place the focus on the development of membranes and graft materials.

Therefore, we completely understand your point since we have not covered all the novel regeneration therapies using the growth factors.

According to your comment, we have clarified the main focus of this literature and put the additional paragraph stating that tissue regeneration therapy using growth factors is also developing. The additional paragraphs are as below(on page 2, 71~72):

“There are many advances in regeneration therapy in periodontal tissues, but only the development of membranes and grafting materials are reviewed in this article.”

Also, please refer to the section on page 4, section 2.2.2 and 2.2.3 since the summary of the membrane releasing growth factors and PRF membrane is contained herein.

Reviewer 2 Report

The manuscript entitled " Past, Present and Future of Regeneration Therapy in Oral and Periodontal Tissue: A Review" presents on a very descriptive way the advances of oral regeneration therapy. The data are well presented, in particular  guided tissue regeneration and guided bone regeneration. However, I feel that there are basic data on the source of cells to be used that is missing, as well as appropriate references to the recent work with these.

It would be nice that some of the steps of membrane development or complex scaffolds or multilayer membranes with a functionally graded structures be depicted schematically on a figure. This would make the review much easier to understand for non-specialists.

Author Response

Response to Reviewer 2 Comments

Point 1:

The manuscript entitled " Past, Present and Future of Regeneration Therapy in Oral and Periodontal Tissue: A Review" presents on a very descriptive way the advances of oral regeneration therapy. The data are well presented, in particular guided tissue regeneration and guided bone regeneration.

However, I feel that there are basic data on the source of cells to be used that is missing, as well as appropriate references to the recent work with these.

It would be nice that some of the steps of membrane development or complex scaffolds or multilayer membranes with a functionally graded structures be depicted schematically on a figure. This would make the review much easier to understand for non­specialists.

Response 1:

Dear reviewer 2.

Thank you for your kind review

We have reviewed a great number of journals on various types of regeneration therapy to write this literature. However, since too many perspectives were discussed in those journals, we have decided to place the focus on the development of membranes and graft materials. Therefore, we do agree that the biologic process regarding regeneration therapy is not sufficiently discussed herein.

If the source of cells mentioned in your comment refers to the cells related to regenerative procedures, please note that it is briefly discussed on page 2, line 54 to 58. As the periodontium is composed of four different tissues, gingival fibroblast, mesenchymal cells originated from the periodontal ligament, osteoblast, osteoclast, and cementoblast are the ones that are related to the regeneration process. Since osteoblast and osteoclast are not thoroughly elucidated in the literature, we have added a more detailed statement to the section of concern.

A variety of journals show structural traits of the membranes through data such as SEM or histologic pictures. We have asked some of the authors for the data, but could not get an answer in the mean period. Therefore, we have instead put a schematic picture showing the three-dimensional structure of the scaffold and its steps of degradation process in figure1 on page 6.

Round 2

Reviewer 2 Report

I still feel that would be nice to have a better description of the different cell possibilities

Author Response

Response to Reviewer 2 Comments

Point 1:

I still feel that would be nice to have a better description of the different cell possibilities

Response 1:

Dear reviewer 2.

Thank you for your kind review

We did not understand exactly what you were asking for. We have added a new technology section to the text that uses cell transplantation in 3D scaffolds. (Highlighted part on page 5) Cells involved in regeneration via GTR and GBR will be seeded on the membrane to facilitate the regeneration process. We have also added four references to this.